# CNNs trained with adult data are useful in pediatrics. A pneumonia classification example

**Maria Rollan-Martinez-Herrera[1], Alejandro A. Díaz[2], Rubén San José Estépar[1], Gonzalo Vegas Sanchez-Ferrero[1], James C. Ross[1], Raúl San José Estépar[1‡]\*, Pietro Nardelli[1‡]**

**1** Department of Radiology, Applied Chest Imaging Laboratory, Harvard Medical School, Brigham and Women's Hospital, Boston, Massachusetts, United States of America, **2** Division of Pulmonary and Critical Care Medicine, Chest Imaging Laboratory, Harvard Medical School, Brigham and Women's Hospital, Boston, Massachusetts, United States of America

‡ RSJE and PN are co-senior authors on this work.
\* rsanjose@bwh.harvard.edu

**Data Availability Statement:** All the data sources used for this study come from open public resources and can be retrieved from the original

## Abstract

### Background and objectives

The scarcity of data for training deep learning models in pediatrics has prompted questions about the feasibility of employing CNNs trained with adult images for pediatric populations. In this work, a pneumonia classification CNN was used as an exploratory example to showcase the adaptability and efficacy of such models in pediatric healthcare settings despite the inherent data constraints.

### Methods

To develop a curated training dataset with reduced biases, 46,947 chest X-ray images from various adult datasets were meticulously selected. Two preprocessing approaches were tried to assess the impact of thoracic segmentation on model attention outside the thoracic area. Evaluation of our approach was carried out on a dataset containing 5,856 chest X-rays of children from 1 to 5 years old.

### Results

An analysis of attention maps indicated that networks trained with thorax segmentation placed less attention on regions outside the thorax, thus eliminating potential bias. The ensuing network exhibited impressive performance when evaluated on an adult dataset, achieving a pneumonia discrimination AUC of 0.95. When tested on a pediatric dataset, the pneumonia discrimination AUC reached 0.82.

### Conclusions

The results of this study show that adult-trained CNNs can be effectively applied to pediatric populations. This could potentially shift focus towards validating adult models over pediatric population instead of training new CNNs with limited pediatric data. To ensure the

source as described in the manuscript. Pediatric dataset: https://data.mendeley.com/datasets/rscbjbr9sj/3 Adult datasets: - NIH dataset: https://arxiv.org/abs/1705.02315 - CheXpert: https://arxiv.org/abs/1901.07031 - PADChest: https://arxiv.org/abs/1901.07441.

**Funding:** This work has been funded by National Institute of Health (NIH) awards R01HL149877 and 5R21LM013670 and a fellowship by Fundación Martin Escudero. The funders had no role in study design, data collection and analysis, decision to publish, or preparation of the manuscript.

**Competing interests:** The authors have declared that no competing interests exist.

generalizability of deep learning models, it is important to implement techniques aimed at minimizing biases, such as image segmentation or low-quality image exclusion.

## Introduction

Deep learning is a promising field that is enabling researchers to understand complex physical and biological phenomena [1]. In recent years, deep learning models have gained relevance in the medical field [2] and many tools are being developed for diagnosis tasks. While the use of deep learning has grown exponentially in adults' imaging, its application in the pediatric field is currently limited. Despite the presence of few papers proposing the use of convolutional neural networks for pediatric imaging [3–5], the scarcity of data, which limits the ability to effectively train neural networks, remains a significant obstacle hindering the wider implementation of deep learning techniques in this specific field. Notably, while different models focusing on pneumonia diagnosis have been proposed in recent years [6–8], the reliance on the same dataset both for training and validation often prevails. Although these models have demonstrated good validation performance, their efficacy is constrained by the inherent limitation of being tested on the same dataset used for training.

This significant and potentially insurmountable limitation raises the question of whether CNNs trained on adult images can be effectively applied to pediatric populations. Considering that model validation typically demands a smaller dataset compared to the extensive data needed for model training, it would be wise to utilize the limited number of available pediatric images to validate well-trained networks originally developed for adult populations. This approach can yield valuable insights and enhance the applicability of these networks in pediatric imaging.

This study is based on the assumption that numerous image classification tasks exhibit similarities between adults and children, with differences that can be effectively addressed through the implementation of techniques aimed at reducing biases and enhancing the generalizability of neural networks. These techniques include meticulous selection of training images or image segmentation [9, 10]. To test this hypothesis, we trained a convolutional neural network using a meticulously curated dataset of adult chest X-ray images. The selection and labeling of these images were performed by multiple experts specifically trained for pneumonia diagnosis. Subsequently, we deployed and validated this network on a publicly available dataset consisting of pediatric chest X-rays images of children from 1 to 5 years old, focusing on the diagnosis of childhood pneumonia.

Our main goal is to investigate the feasibility of applying models trained on adult images to pediatric populations. The novelty of our works lies in the adaptation and validation of a model trained on adult x-ray images for the classification of pediatric conditions, demonstrating its potential to overcome the scarcity of pediatric training data. By doing so, we aim to lay the groundwork for future deep learning projects in pediatrics. Specifically, models trained with adult images can have an impact across various domains within pediatric medicine, such as MRI intracranial lesions, otitis in otoscopy, dermatological infections, or electrocardiogram.

Our approach focuses on validating adult-trained models on pediatric datasets, a strategy that requires fewer images compared to training models solely with pediatric data. This approach has the potential to significantly advance deep learning applications in pediatric healthcare. However, it is essential to acknowledge that pediatric pathology can diverge significantly from adult pathology, potentially limiting the generalizability of this approach in certain scenarios.

Table 1. Dataset used in the experiment.

| Dataset | Source | Sample | Folders | Task |
| --- | --- | --- | --- | --- |
| **Adult Dataset** | NIH | Selection (46947) | Training 67%. Testing 33% | Pneumonia Classification |
| | CheXpert | | | |
| | PadChest | | | |
| **Pediatric Dataset** | Chest X-Ray Pneumonia | All images (5856) | Testing 100% | Pediatric Validation |

## Material and methods

### Data

In this retrospective study, we conducted analyses on a de-identified dataset comprising both adult and pediatric frontal chest X-rays (CXR) Institutional review board (IRB) approval was waived. Table 1 summarizes the different datasets used in this study. Prior to our access and use, these datasets had undergone thorough de-identification processes to ensure the anonymity and privacy of individuals whose images were included. This de-identification process involved the removal of all personal health information that could be used to identify the subjects, in compliance with HIPAA Privacy Rules. The retrospective pediatric cohort consisted of participants from one to five years old from Guangzhou Women and Children's Medical Center, Guangzhou. All chest X-ray imaging was performed as part of patients' routine clinical care. Institutional Review Board (IRB)/Ethics Committee approvals were obtained by the primary study [11].

Given the retrospective nature of the study and the use of de-identified data, the requirement for Institutional Review Board (IRB) approval was waived. This waiver was in accordance with ethical guidelines for research involving human subjects, as the study posed minimal risk to participants due to the lack of identifiable information. Furthermore, the use of existing, publicly available datasets that are non-identifiable aligns with the exemption criteria for IRB review under federal regulations. Our research adhered strictly to the principles of ethical conduct in medical research, ensuring the protection of individual privacy while contributing to scientific understanding and advancement.

The **Adult Dataset** consists of data extracted from three different datasets: NIH ChestX-ray8, containing 112,120 chest X-ray images from 30,805 adult patients (https://arxiv.org/abs/1705.02315); CheXpert, comprising 224,316 chest X-ray images from 65,240 adults (https://arxiv.org/abs/1901.07031); and PadChest consisting of 160,000 chest X-ray images from 67,000 adult subjects (https://arxiv.org/abs/1901.07441). From these images, some were excluded. Specifically, images presenting two or more external devices (cables, leads, tubes, pacemakers...) over the lung field were removed. Additionally, images exhibiting extreme angular rotations, positioning, or with incorrect pulmonary fields (image truncation) were omitted.

All images originally diagnosed with pneumonia were reviewed and classified by five trained readers. Another subset of images, originally labelled as healthy, were also reviewed to ensure to ensure the accuracy of their classification. Label consensus for all images was reached using the Dawid-Skene approach [12].

Finally, the selected images were separated into a training dataset of 32,088 images and a testing dataset consisting of 14,859 cases. Proportions for this testing dataset were as follows: 47% of images represented healthy cases and 63% depicted pneumonia.

For validation on pediatric data, a publicly available dataset of pediatric images (**Pediatric Dataset,** https://data.mendeley.com/datasets/rscbjbr9sj/3) was used. This dataset contains a total of 5,856 anterior-posterior chest X-ray images from retrospective cohorts of pediatric

patients of one to five years old from Guangzhou Women and Children's Medical Center, Guangzhou. Chest X-ray images were obtained as part of patients' routine clinical care and were screened for quality control and diagnosis by three expert physicians.

The images in the original dataset were categorized into separate subsets for training, validation, and testing. However, for our experiments we combined these subsets together. Among these images, 4,273 cases were diagnosed with pneumonia, with 1,493 classified as viral pneumonia and 2,780 as bacterial pneumonia. The remaining 1,583 subjects within the dataset exhibit no sign of pneumonia.

## Preprocessing

All images underwent a series of preprocessing operations to enhance their quality, using the Python OpenCV library, models were developed in Tensorflow 2.15. First, local contrast enhancement was applied using adapting histogram equalization. Then, median filtering was applied to reduce noise, and global contrast stretching was implemented through percentile adjustment to further enhance contrast. As a final step, all images were reshaped to 512 x 512 pixels and a z-score normalization was applied to ensure consistent and comparable pixel values across all images.

## Neural network architecture

In this study, different neural network architectures that have been previously proposed in the literature were considered [13–15]. Hyperparameter optimization was conducted using Mango [16], exploring three different backbone architectures: Xception [13], Inception ResNet [14], and EfficientNet 2 [15]. We adjusted learning rates, frozen proportions, and thorax segmentation methods as part of this optimization process. This exploration involved 139 unique combinations of hyperparameters and backbones architectures, was repeated three times for robustness, and was validated using mean F1 score. Through this process, the Xception architecture was chosen, as it demonstrated the highest mean F1 score among the evaluated models.

In addition, we evaluated two training approaches: initializing the model with ImageNet weights and training it from scratch. As expected, leveraging pretrained ImageNet weights resulted in enhanced performance. This strategy not only offers computational efficiency by reducing training time and computational resources but also improves effectiveness through feature reuse and generalization [17].

While X-ray images typically consist of only one channel, the Xception network requires a three-channel image as input. To address this, two options were considered: the first involved duplicating the single channel image three times, but it yielded no additional learning. The second option, employing an initial 2D convolutional layer, was preferred despite its increased computational complexity, as it showed enhanced learning and model performance. Additionally, the last fully connected layer of the original network was replaced with a global max pooling layer, followed by four dense layers and two dropout layers. Finally, the network was designed to produce two outputs: healthy and pneumonia. Fig 1 shows the network architecture used in this work and a block diagram outlining the training and validation processes.

## Neural network training

As part of the network selection, the impact of thorax segmentation on network performance was also assessed [18]. To this end, training was conducted both with and without segmentation and results were evaluated using a GradCAM approach [19]. This approach was employed to analyze the attention distribution of the network beyond the thorax region, enabling the

## Model architecture

## Workflow

**Fig 1. Network architecture and workflow.** Our network architecture (left) follows the structure of CNN with an Xception backbone to extract features and a fully connected stage. Our training and validation workflow (right) exploits three curated and preprocessed databases. The model was trained on the adult dataset, and then we carried out independent validation on the adult and pediatric datasets. Further analysis was performed in the pediatric dataset for model explainability.

examination of potential biases and ensuring a comprehensive evaluation of the network's performance.

The network was implemented in Tensorflow [20] and hyperparameters were optimized using the Mango approach [16] to fine-tune the network's configuration settings. After hyperparameters selection, the network was trained with an Adam optimizer for 200 epochs (with an early stopping mechanism of 5 epochs) at a learning rate of $1e^{-4}$, batch size of 8, and a

categorical cross-entropy as loss function. Moreover, a randomized 70–30% split was used for training and validation.

The Xception backbone of the network was partially frozen during training, with the initial portion of the network keeping the ImageNet weights fixed, while the remaining portion underwent fine-tuning. By means of a hyperparameter tuning process, we explored a wide spectrum of frozen proportions, ranging from a full training of the model from scratch to nearly complete freezing of the backbone. After meticulous analysis of the hyperparameter results, it was determined that fine-tuning the model represented the most effective approach, showing substantial enhancements. Weights were kept fixed in 40% of the network architecture when training with thorax segmentation and in 50% of the layers when no segmentation was used.

### Neural network validation

The neural network was evaluated using both the Adult Dataset and the Pediatric Dataset to assess its performance. The Adult Dataset and Pediatric dataset were used to analyze the network's capability in discriminating healthy and pneumonia individuals.

### Interpretability of the neural network's behavior

To thoroughly analyze the behavior of the network trained with and without thorax segmentation and to discern potential biases, we employed the GradCAM visualization technique [19]. The goal was to assess if the network's percentage of attention outside the thorax region decreased when including thorax segmentation to the training, helping the network classification performance.

To achieve this, the heatmap extracted from the image by GradCAM was binarized using a threshold value of 0.1. The determination of this specific threshold underwent meticulous experimentation and rigorous testing to ensure its appropriateness for the task at hand. Through a series of systematic experiments and multiple iterations, we conducted thorough analyses to identify the optimal threshold value. This involved visually assessing various performance results across different threshold values to ascertain the threshold that maximizes the desired outcome. By quantifying the number of corresponding pixels between the binarized heatmap and the thorax region, we identified the network's attention points within the thorax. Conversely, the attention pixels from the binarized heatmap that did not align with the thorax region represented the network's attention points outside the thorax.

To quantify the network's attention outside the thorax we measured the ratio between the attention pixels outside the thorax and the total number of pixels outside the thorax. This helped us to understand how much attention each model was allocating outside the relevant thoracic region. This method was applied over the Pediatric Dataset.

### Statistics

The network's performance was evaluated using Accuracy, Precision, Recall, F1 score and ROC curve analysis. AUC 95% confidence intervals were estimated using bootstrapping. To evaluate the differences in the attention of the network outside of the thorax region Wilcoxon test was used with $p < 0.05$ as cut-off point for statistical significance.

## Results
### Neural network interpretability results

In this section, we report the attention results obtained when training the network with and without thorax segmentation, to study the effects of the segmentation on CNN's attention areas. The median proportion of attention of the network outside the thorax region, relative to

the entire area outside the thorax, was 8.69% (IQR 4.08–14.20%) for training without segmentation and 2.31% (IQR 1.21–3.97%) when the segmentation was included. These proportions yielded significant difference between the two networks ($p < 0.0001$).

The network's attention outside the thorax region was observed to be higher in pneumonia images compared to healthy images when training the network both without (9.17% vs 7.76%, $p < 0.0001$) and with (2.32% vs 2.31%, $p = 0.007$) thorax segmentation. Similarly, when comparing attention outside the thorax in bacterial images in comparison to viral images, it was found to be higher in bacterial images when training both without (9.76% vs 8.02%, $p < 0.0001$) and with (2.4% vs 2.25%, $p = 0.03$) segmentation. On the other hand, no differences in attention between viral images and healthy cases were found for the network trained with thorax segmentation (2.31% vs 2.25%, $p = 0.45$), whereas statistically significant differences were present for the network trained without the segmentation (8.02% vs 7.76%, $p < 0.0008$). Fig 2 shows examples of attention heatmaps to provide a visual illustration.

## Neural network validation

Based on the attention analysis results detailed in the previous section, training was performed using thorax segmentation to help address biases that may originate from the network focusing

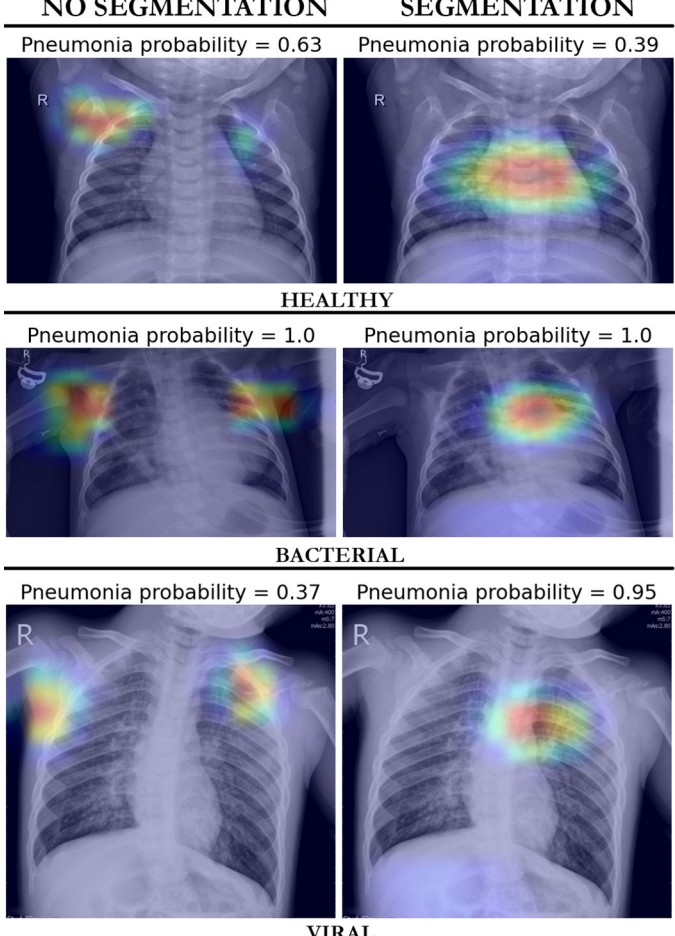

**Fig 2. Heatmaps and corresponding pneumonia probabilities.** Images were obtained when training the network with and without thorax segmentation.

# Pneumonia

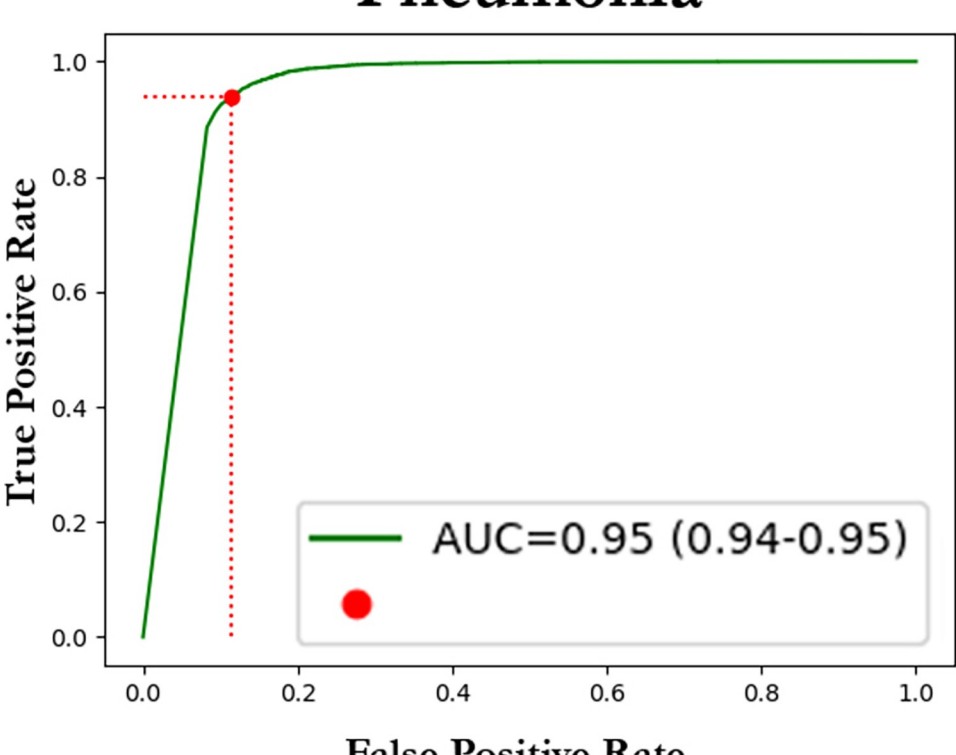

**Fig 3. ROC plot for the adult dataset.** ROC curve depicting the performance of our model for pneumonia discrimination for the Adult Dataset. An AUC of 0.95 (95%CI: 0.94–0.95) was obtained, showing the good performance of the presented network.

outside the thoracic region. The network's performance was evaluated on the testing subsets of the Adult Dataset (Fig 3) and on the Pediatric Dataset (Fig 4). In Fig 5, some examples of pneumonia prediction of images extracted from the Pediatric Dataset are shown.

A summary of the outcomes and performance metrics for the results in pneumonia discrimination can be found in Table 2.

## Discussion

The primary objective of this work was to demonstrate the potential applicability of CNNs trained with adult images in the context of pediatric imaging. To this end, we conducted a comprehensive experiment where a CNN was trained specifically for pneumonia diagnosis using adult chest X-ray images, while carefully minimizing bias. Once trained, we evaluated the performance of the network on a dataset of X-ray images of children from 1 to 5 years old.

To ensure the reliability of our findings and mitigate bias as well as interobserver variability, we implemented a meticulous approach in the selection and labeling of the training adult images by a group of experts. This collective effort enhanced the accuracy and robustness of our dataset. Additionally, in order to investigate the potential benefits of thorax segmentation to reduce bias when applying models to a considerably distinct dataset, we trained the neural network both with and without thorax segmentation. This allowed us to examine the impact of the segmentation on the behavior of the neural network, determining the specific areas it prioritizes in both cases. Through this analysis, we were able to identify the neural network with

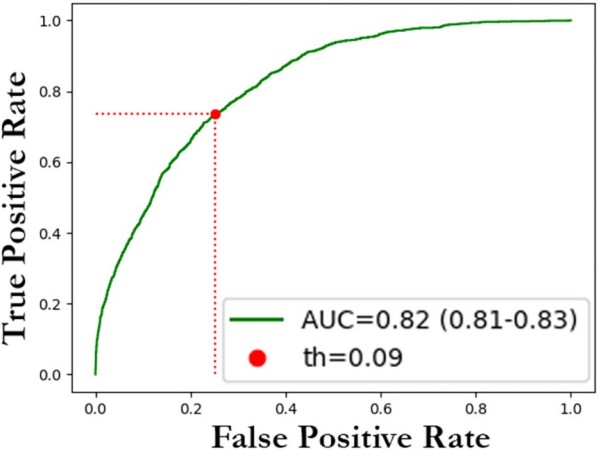

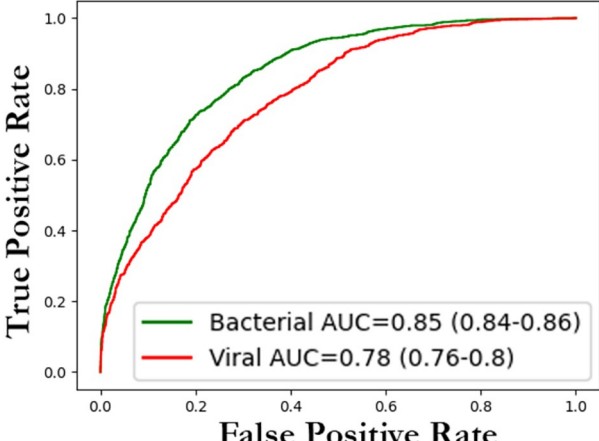

**Fig 4. ROC plots for the pediatric dataset.** ROC curve for pneumonia discrimination was obtained using all images from the Pediatric Dataset. Although the images in this dataset were distinct from the training dataset, our model achieved an AUC of 0.82 (95%CI: 0.8–0.83), highlighting the model's promising performance in accurately distinguishing pneumonia cases within pediatric populations.

the best performance, showcasing its effectiveness in the given task. To this end, the network was evaluated on both the Adult Dataset and the Pediatric Dataset.

In particular, our results show that thorax segmentation plays a crucial role in directing the network's attention predominantly towards the thorax region. An interesting finding is that the differences in attention outside the thorax between different groups of images (bacterial, viral and normal) are less pronounced in the network trained with thorax segmentation compared to the network trained without the segmentation. These findings suggest that the use of thorax segmentation not only reduces the attention outside the thorax region but also diminishes the disparities in attention allocation observed across different pathologies. These findings emphasize the significant benefits of incorporating thorax segmentation for enhancing the network's focus and minimizing attention biases in the analysis of pediatric chest X-ray images.

Testing results on the Adult Dataset were remarkable (Table 2, Fig 3). Our Receiver Operating Characteristic (ROC) plot showcases the efficacy of our model in discriminating pneumonia cases within the Adult Dataset. With an AUC score of 0.95, accompanied by a confidence interval (CI) ranging from 0.94 to 0.95, our network demonstrates strong performance in

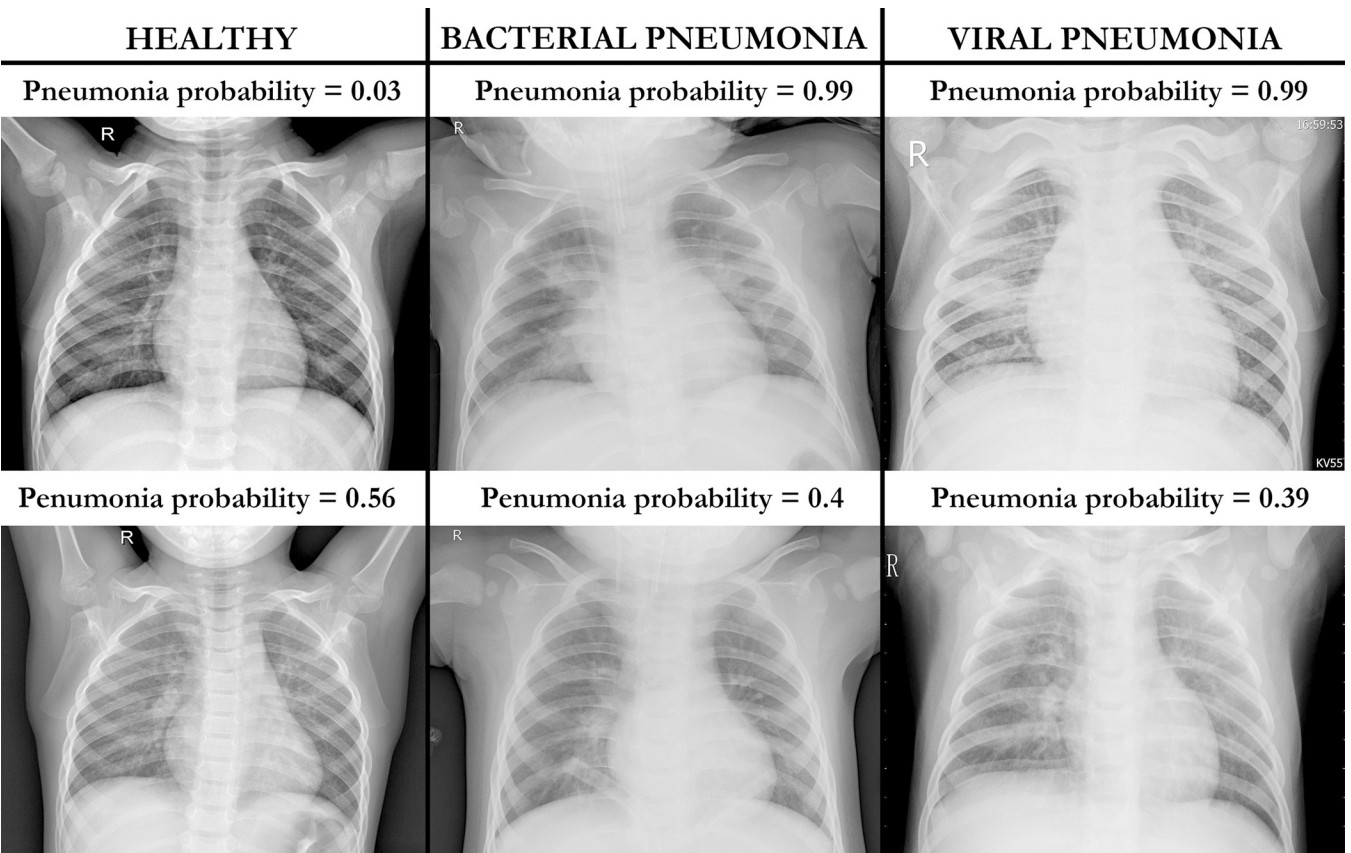

**Fig 5. Examples of pneumonia predictions on images from the pediatric dataset.** Images corresponding to the Pediatric dataset were provided by the proposed neural network. Upper images reflect areas where the network provided accurate predictions based on the network probabilities. Lower images reflect cases where the network was less confident about the decision. (Left) We observe two clear chest X-ray images. The upper image exhibits lung fields without abnormalities, while the lower one, although lacking signs of pneumonia, displays some bilateral perihilar infiltrates, thus, potentially explaining why our proposed networks give a mid-range probability value despite being a healthy case. (Middle) These two chest X-rays indicate bacterial pneumonia, typically more severe than viral pneumonia. The upper image displays pronounced bilateral infiltrates, particularly notable in the right hemithorax, whereas the lower image exhibits subtle right paramilitary infiltrates, less pronounced than the upper ones, explaining the lower confidence of the network as reflected by the predicted probability. (Right) These two cases illustrate viral pneumonia, which typically spreads more diffusely across both lung fields. The upper image reveals a bilateral diffuse infiltrate, with a more pronounced presence in the right hemithorax, while the lower image displays a similar pattern but with lesser intensity. The images in the lower row, despite their difference in disease status, reveal noticeable radiographic similarities with subtle paramilitary infiltrates without distinct consolidations, explaining why they share similar prediction probabilities.

distinguishing between pneumonia-positive and pneumonia-negative instances. These findings underscore the reliability and potential clinical utility of our presented network.

Regarding the applicability of the neural networks trained with adult images to pediatric CXR, we observed a promising AUC value of 0.82 (CI 95% 0.81–0.83) (Fig 4). These results are

**Table 2. Validation results for pneumonia discrimination of the neural network (trained with adult data) on both the adult and the pediatric datasets.**

|  | Adult Dataset (CI 95%) | Pediatric Dataset (CI 95%) |
|---|---|---|
| **AUC** | 0.95 (0.94–0.95) | 0.82 (0.81–0.83) |
| **F1 score** | 0.82 (0.81–0.83) | 0.76 (0.75–0.76) |
| **Precision** | 0.87 (0.86–0.87) | 0.79 (0.78–0.8) |
| **Accuracy** | 0.82 (0.81–0.83) | 0.75 (0.75–0.76) |
| **Recall** | 0.82 (0.81–0.83) | 0.75 (0.75–0.76) |

particularly encouraging, considering that this validation was conducted on a dataset containing patient types that differ from those included in the training dataset.

The meticulous preselection of training images, specifically excluding those featuring external devices or exhibiting poor quality, combined with the thorax segmentation approach, is likely to have positively impacted the model's generalizability that we observe.

Training a neural network directly on a dataset of pediatric images would likely yield better results, especially if validated using a dataset with similar characteristics. Previous studies [4–8], some of which used the same pediatric dataset utilized here, have followed a similar approach. These studies achieved metrics above 0.9, outperforming the presented method. Notably, all of these models used the same dataset containing pediatric chest X-ray images for both training and validation, explaining their superior results. While we would aim to achieve comparable performance, we recognize the challenge inherent in achieving such accuracy with our approach.

It is essential to acknowledge the limited generalizability of the previous approaches, as its performance on datasets with different characteristics remains uncertain. In contrast, our study highlights the robustness of our method across diverse types of data. Moreover, we want to point out that the primary goal of this study was not to develop the best performing model, but rather to demonstrate the potential applicability of CNNs trained on adult images for pediatric image analysis. Additional fine-tuning of adult-based models with pediatric datasets could improve the results presented in this work.

In this work, we decided to use a pre-trained model based on ImageNet. Transfer learning based on pre-trained models provides a robust foundational knowledge of visual concepts, which typically leads to enhanced performance across a variety of downstream tasks, including object recognition, image classification, and more complex tasks such as scene understanding. Furthermore, while our dataset is sizable, it does not reach the scale of millions of images often used in other contexts, making transfer learning a more efficient strategy. This method allows the model to leverage the generic features learned from ImageNet and adapt them to the specific characteristics of our task. Additionally, pre-training a model on ImageNet can enhance its robustness and ability to generalize to different populations as intended in this work.

While our findings are promising, further studies and extensive validation are required to demonstrate a full applicability of CNNs trained with adult data to pediatric populations. This study highlights only a single instance of their usefulness in diagnosing pneumonia. However, it would be highly valuable to explore additional examples across various radiological domains and even other medical specialties, such as dermatology or cytology. By evaluating these diverse scenarios, we can better ascertain the generalizability and effectiveness of CNNs in pediatric healthcare. This will provide valuable insights and contribute to the broader adoption of CNNs in pediatric medical practice.

Another major limitation of this study is that we did not have access to demographic characteristics of the pediatric data sources that were used for this study. Therefore, we could exclude biases in the model related to gender and/or age. The importance of implementing strategies that can effectively minimize biases and maximize the generalizability of deep learning models should not be underestimated. Meticulous selection of training images and the incorporation of techniques such as image segmentation play pivotal roles in achieving these objectives.

## Conclusions

The field of pediatrics has faced challenges in leveraging deep learning advancements, primarily due to the limited availability of large-scale pediatric population cohorts. However, there is

potential in adapting technologies from adult populations to develop child-specific solutions, offering a promising avenue for progress.

Our experiments yielded highly satisfactory and encouraging results. Training a pneumonia neural network with adult images showcased a promising performance in pediatric cases, highlighting the potential of using CNNs trained with adults' data for applications in pediatrics. These groundbreaking findings have the potential to revolutionize deep learning research by shifting focus from solely training new deep learning models with scarce pediatric images to use this small data for validating networks trained with abundant adult images, which are more readily available, for pediatric cases. This shift in approach not only capitalizes the abundance of adult data but also addresses the scarcity of pediatric data, thereby expanding the possibilities and advancements in pediatric deep learning research.

## Author Contributions

**Conceptualization:** Maria Rollan-Martinez-Herrera, Raúl San José Estépar, Pietro Nardelli.

**Data curation:** Maria Rollan-Martinez-Herrera, Alejandro A. Díaz, Rubén San José Estépar, Gonzalo Vegas Sanchez-Ferrero, James C. Ross, Raúl San José Estépar, Pietro Nardelli.

**Formal analysis:** Maria Rollan-Martinez-Herrera.

**Supervision:** Raúl San José Estépar.

**Validation:** Maria Rollan-Martinez-Herrera.

**Writing – original draft:** Maria Rollan-Martinez-Herrera.

**Writing – review & editing:** Maria Rollan-Martinez-Herrera, Alejandro A. Díaz, Rubén San José Estépar, Gonzalo Vegas Sanchez-Ferrero, James C. Ross, Raúl San José Estépar, Pietro Nardelli.

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
