## [Decision Letter · Decision Letter 0]

12 Mar 2024

PONE-D-24-00173Adult-Trained Deep Learning in Pediatric Pneumonia Diagnosis.PLOS ONE

Dear Dr. San José Estépar,

Thank you for submitting your manuscript to PLOS ONE. After careful consideration, we feel that it has merit but does not fully meet PLOS ONE’s publication criteria as it currently stands. Therefore, we invite you to submit a revised version of the manuscript that addresses the points raised during the review process.

We look forward to receiving your revised manuscript.

Kind regards,

Lorenzo Faggioni, M.D., Ph.D.

Academic Editor

PLOS ONE

Journal Requirements:

"This work has been funded by NIH awards R01HL149877 and 5R21LM013670 and a fellowship by Fundación Martin Escudero"

3. Please expand the acronym “NIH” (as indicated in your financial disclosure) so that it states the name of your funders in full.

Reviewers' comments:

Reviewer's Responses to Questions

**Comments to the Author**

1. Is the manuscript technically sound, and do the data support the conclusions?

Reviewer #1: Partly

Reviewer #2: Yes

2. Has the statistical analysis been performed appropriately and rigorously? 

Reviewer #1: I Don't Know

Reviewer #2: Yes

3. Have the authors made all data underlying the findings in their manuscript fully available?

Reviewer #1: Yes

Reviewer #2: Yes

4. Is the manuscript presented in an intelligible fashion and written in standard English?

Reviewer #1: Yes

Reviewer #2: Yes

5. Review Comments to the Author

Reviewer #1: In the study, the authors tried to classify pediatric x-ray images training the model with adult x-ray images. They used the Xception deep learning model as the method. There are some deficiencies in the article and these deficiencies must be completed before the article can be published.

1) There are many current publications in the literature regarding the classification of pediatric x-ray images as pneumonia or not, but these are not mentioned in the article. The reference list is not up to date.

2) They used Xception as a model, so there is no innovation in the article in terms of method.

3) It would be appropriate to add the architecture used to the article as a figure and show what has been done as a block diagram.

4) While training, the authors trained the model in 3 classes (healthy, mild pneumonia, moderate-severe pneumonia) using adult x-ray images, but when testing (validating), they used pediatric images and tested it in 2 classes (pneumonia and healthy). However, looking at Figure 3, the classes in the ROC curve are bacterial pneumonia and viral pneumonia, there is an inconsistency. This needs to be fixed.

Reviewer #2: The manuscript investigated the feasibility of employing CNNs trained with adult images for pediatric populations. In general, the manuscript is well presented. The model performance is not excellent and lacks benchmarking. Please see below for my detailed comments.

Title:

1. The “Adult-Trained” in the title might be a little misleading. I think it is better to specify the model was trained with adult data.

Abstract:

2. Line 28: “Background and Objectives: The scarcity of data for training deep learning models in pediatrics…” In general, this is true in the area of pediatric study. However, the authors were able to curate 5,856 chest X-rays of children, which is a pretty large dataset for medical imaging study. This seems contradicting to me.

3. line 39: “achieving a macro AUC of 0.9 and a pneumonia discrimination AUC of 0.95.” Please define the “macro AUC” here.

Introductions:

4. Please briefly talk about if there are any other deep learning studies on the diagnosis of childhood pneumonia. If so, what are their model performances and how the current study distinguished itself from them.

5. Line 75: “By doing so, we aim to lay the groundwork for future deep learning projects in pediatrics.” There are many different areas in pediatrics. In some of them, adult disease and pediatric disease can be very different. So I think the authors need to be specific on this statement.

Methods

6. In the Preprocessing section, please indicate the programs and packages that were used.

7. Line 150: “After extensive testing, the Xception architecture, initialized with ImageNet weights, was chosen and customized as a backbone due to its reliability.” Please specify what tastings were performed for this model selection process.

8. Did the authors try to train the model from scratch instead of using ImageNet weights? Since the training dataset is quite large (n=46,947) and the classification task is fairly easy, training from scratch should be able to achieve a pretty good model performance. Additionally, manually broadcast the data from 1 channel to 3 channels for such a large dataset adds a lot of computation for model training.

9. Line 193: “To achieve this, the heatmap extracted from the image by GradCAM was binarized using a threshold value of 0.1.” Are there any particular reasons to choose this threshold?

Results:

10. Please provide a brief description on the patient characteristics.

11. Please provide a 95% confidence interval for the model evaluation metrics in both Results section and Table 2.

12. I am curious why the authors did not use a transfer learning approach that uses part of the pediatric data to fine tune the adult model. The AUC (0.82) and F1-score (0.76) are not excellent for a binary classification task. Transfer learning should further boost up the model performance.

13. It would be helpful to benchmark the current model with a model trained based on the pediatric data.

14. Since the primary objective of the current study is to apply a model trained from adult data to predict on pediatric data, what is the point to perform 3-class classification on adult model while the pediatric data only has two classes?

Discussion and conclusion:

15. The authors should also discuss the current model performance with other similar studies.

6. PLOS authors have the option to publish the peer review history of their article (what does this mean?). If published, this will include your full peer review and any attached files.

Reviewer #1: No

Reviewer #2: No

---

## [Author Response · Author response to Decision Letter 0]

29 May 2024

Dear Editor,

We thank you for the opportunity to revise our manuscript and for the constructive feedback provided by the reviewers. Their insightful comments and suggestions have been invaluable in guiding us to enhance the quality and clarity of our work. We have carefully addressed each point raised by the reviewers, and these revisions have significantly improved our manuscript, making it a stronger contribution to the field. We appreciate the reviewers' thorough evaluations and the chance to refine our study based on their expert recommendations.

Thank you for considering our manuscript for publication in PLOS One. We look forward to the possibility of contributing to the scientific community through your esteemed journal.

Response to Reviewer #1

Response to Comment 1: “There are many current publications in the literature regarding the classification of pediatric x-ray images as pneumonia or not, but these are not mentioned in the article.”

We appreciate your insightful observation regarding the literature on pediatric X-ray image classification. We acknowledge the omission and have updated our reference list to include current publications relevant to pediatric pneumonia classification using deep learning models. These additional references provide a broader context and reinforce the novelty and significance of our approach in employing adult X-ray images for pediatric classification tasks. The revised manuscript now features a comprehensive review of recent advancements in the field, highlighting our study's position within this emerging research landscape.

Response to Comment 2: “They used Xception as a model, so there is no innovation in the article in terms of method.”

While we understand the concern regarding the perceived lack of methodological innovation due to our use of the Xception model, our study introduces a novel application of this established model to address a specific challenge in pediatric imaging. The innovative aspect of our work lies in the adaptation and validation of a model trained on adult x-ray images for the classification of pediatric conditions, demonstrating its potential to overcome the scarcity of pediatric training data. This approach not only underscores the versatility of the Xception model, but also contributes to the growing body of research on cross-age applicability of deep learning models in medical imaging.

Response to Comment 3: “It would be appropriate to add the architecture used to the article as a figure and show what has been done as a block diagram.”

In response to your suggestion, we have added a detailed figure depicting the architecture of the Xception model as used in our study, along with a block diagram that outlines the training and validation processes. This visual enhancement clarifies the methodological framework and provides readers with a clear understanding of the workflow and modifications applied to adapt the model for pediatric classification tasks.

Response to Comment 4: “While training, the authors trained the model in 3 classes (healthy, mild pneumonia, moderate-severe pneumonia) using adult x-ray images, but when testing (validating), they used pediatric images and tested it in 2 classes (pneumonia and healthy). However, looking at Figure 3, the classes in the ROC curve are bacterial pneumonia and viral pneumonia, there is an inconsistency. This needs to be fixed.”

We acknowledge the inconsistency identified in the representation of the classes in Figure 3 and have rectified this error. The revised figure accurately reflects the testing (validation) classes, aligning with the description in the text. We have also elaborated on the rationale behind the classification schema used during training and testing, providing clarity on the methodological decisions and ensuring consistency across the manuscript.

Response to Reviewer #2

Title Response: 

1. “The “Adult-Trained” in the title might be a little misleading. I think it is better to specify the model was trained with adult data.”

Thank you for pointing out the potential ambiguity in the title. We have revised it to add clarity and specify that the model was trained using adult data, properly reflecting the study's methodology.

Abstract Responses: 

2. “Line 28: “Background and Objectives: The scarcity of data for training deep learning models in pediatrics…” In general, this is true in the area of pediatric study. However, the authors were able to curate 5,856 chest X-rays of children, which is a pretty large dataset for medical imaging study. This seems contradicting to me.”

We acknowledge the apparent contradiction and have revised the statement to better reflect the context in which the large pediatric dataset was curated and its significance within the scope of our study. The revised abstract now accurately conveys the challenge of data scarcity in pediatric studies and the relevance of our large dataset in addressing this issue. Although our pediatric dataset has an adequate sample size for validation, the sample size for training could be much higher, considering the size of the datasets used in X-ray imaging diagnosis using AI.

3. “Line 39: “achieving a macro AUC of 0.9 and a pneumonia discrimination AUC of 0.95.” Please define the “macro AUC” here.

Thanks for bringing this to our attention. In alignment with comment 4 from reviewer 1, we have removed the macro AUC metric from our analysis, as we are now focusing on presenting discrimination results between pneumonia vs healthy classifications.

Introduction Response:

4. “Please briefly talk about if there are any other deep learning studies on the diagnosis of childhood pneumonia. If so, what are their model performances and how the current study distinguished itself from them.”

We have included a brief overview of existing deep learning studies on the diagnosis of childhood pneumonia in the introduction, comparing their model performances with ours in the paper’s discussion. This addition not only situates our work within the existing literature but also highlights the distinctiveness and contribution of our study to the field.

5. Line 75: “By doing so, we aim to lay the groundwork for future deep learning projects in pediatrics.” There are many different areas in pediatrics. In some of them, adult disease and pediatric disease can be very different. So I think the authors need to be specific on this statement.

Thank you for this insightful feedback. We appreciate the point regarding the specificity of our statement regarding the application of deep learning in pediatrics.

It is true that there is vast diversity within pediatric medicine, where diseases can manifest differently in children compared to adults. To address this concern, we have provided a more detailed delineation of the areas within pediatrics where deep learning can have a significant impact. By identifying specific pediatric diseases or conditions, we highlighted the unique challenges and opportunities for applying deep learning techniques effectively in pediatric healthcare.

Methods Responses:

6. In the Preprocessing section, please indicate the programs and packages that were used.

We have specified that the Python OpenCV library was used in the preprocessing stage and the Tensorflow library that we used for creating, training and validating the CNN model, offering transparency and reproducibility of our methodology.

7. Line 150: “After extensive testing, the Xception architecture, initialized with ImageNet weights, was chosen and customized as a backbone due to its reliability.” Please specify what tastings were performed for this model selection process.

The model selection process is now explicitly described, including the criteria and comparisons made to justify the choice of Xception architecture and the use of ImageNet weights.

8. Did the authors try to train the model from scratch instead of using ImageNet weights? Since the training dataset is quite large (n=46,947) and the classification task is fairly easy, training from scratch should be able to achieve a pretty good model performance. Additionally, manually broadcast the data from 1 channel to 3 channels for such a large dataset adds a lot of computation for model training.

We have not pursued training the model from scratch as suggested by the reviewer. Employing transfer learning techniques through fine-tuning pre-trained models is a common practice in the field, rather than initializing a model from scratch. Pre-trained models provide a robust foundational knowledge of visual concepts, which typically leads to enhanced performance across a variety of downstream tasks, including object recognition, image classification, and more complex tasks such as scene understanding. This approach is particularly advantageous in scenarios where collecting a large amount of labeled data is either costly or impractical.

Furthermore, while our dataset is sizable, it does not reach the scale of millions of images often used in other contexts, making transfer learning a more efficient strategy. This method allows the model to leverage the generic features learned from ImageNet and adapt them to the specific characteristics of our task. Additionally, pre-training a model on ImageNet can enhance its robustness and generalization ability. Models trained on diverse and large datasets like ImageNet typically exhibit greater robustness and superior generalization to new, unseen data than those trained on more homogeneous or smaller datasets. This aspect is particularly critical in our work, where we aim to adapt methodologies developed for adult populations to pediatric applications.

We have expanded our discussion to incorporate the important point raised by the reviewer.

9. Line 193: “To achieve this, the heatmap extracted from the image by GradCAM was binarized using a threshold value of 0.1.” Are there any particular reasons to choose this threshold?

The rationale behind the chosen threshold for binarizing the heatmap from GradCAM is now provided, detailing the methodological considerations that guided this decision.

Results Responses:

10. Please provide a brief description on the patient characteristics

Unfortunately, patient characteristics are not available for the public dataset used in this study. We have included this lack of demographic information as a limitation of the data sources that we have employed.

11. Please provide a 95% confidence interval for the model evaluation metrics in both Results section and Table 2.

We have included the 95% confidence intervals for model evaluation metrics in both the Results section and Table 2, offering a more comprehensive statistical representation of our model's performance.

12. I am curious why the authors did not use a transfer learning approach that uses part of the pediatric data to fine tune the adult model. The AUC (0.82) and F1-score (0.76) are not excellent for a binary classification task. Transfer learning should further boost up the model performance.

We appreciate your insightful feedback. In our study, we intentionally focused on evaluating a model trained exclusively on adult data when applied to pediatric cases. This approach was chosen to establish a robust benchmark for future research endeavors and to explore the feasibility of utilizing existing adult-trained models in pediatric contexts.

By abstaining from incorporating pediatric data during training, we aimed to test the model's generalization capabilities under challenging conditions. We agree that transfer learning could enhance model performance, and we acknowledge its potential value in future investigations. However, our study underscores the viability of leveraging adult-trained models for pediatric applications, providing a valuable avenue for validating existing models in pediatric settings and using adult pretrained models as a starting point for further development in pediatric applications.

13. It would be helpful to benchmark the current model with a model trained based on the pediatric data.

Thank you for your suggestion. We acknowledge the potential value in benchmarking our current model against others trained specifically on pediatric data, as it could offer valuable insights. However, the limited number of cases available makes this training impractical in a reliable and robust way. We have acknowledged this fact in the discussion and leave it as future work to compare models trained on adult vs. pediatric-specific models as a reasonable validation study if pediatric datasets are available. However, this validation study now falls outside this paper's intended scope.

14. Since the primary objective of the current study is to apply a model trained from adult data to predict on pediatric data, what is the point to perform 3-class classification on adult model while the pediatric data only has two classes?

After careful analysis, we have decided to remove the 3-class classification on adult data to maintain coherence with the approach adopted for pediatric images.

Discussion and Conclusion Response:

15. The authors should also discuss the current model performance with other similar studies.

We have expanded the discussion on our model's performance compared to other studies, contextualizing our results within the broader landscape of deep learning applications in pediatric pneumonia diagnosis. This comparison highlights our study's contributions and outlines potential avenues for future research and improvement.

We hope these revisions address the concerns raised and further strengthen our work's contribution to the field. We are grateful for the opportunity to clarify these aspects and believe that the amendments significantly enhance the manuscript's quality and impact.

---

## [Decision Letter · Decision Letter 1]

23 Jun 2024

CNNs trained with adult data are useful in pediatrics. A pneumonia classification example.

PONE-D-24-00173R1

Dear Dr. San José Estépar,

We’re pleased to inform you that your manuscript has been judged scientifically suitable for publication and will be formally accepted for publication once it meets all outstanding technical requirements.

Kind regards,

Lorenzo Faggioni, M.D., Ph.D.

Academic Editor

PLOS ONE

Reviewers' comments:

Reviewer's Responses to Questions

**Comments to the Author**

1. If the authors have adequately addressed your comments raised in a previous round of review and you feel that this manuscript is now acceptable for publication, you may indicate that here to bypass the “Comments to the Author” section, enter your conflict of interest statement in the “Confidential to Editor” section, and submit your "Accept" recommendation.

Reviewer #1: All comments have been addressed

Reviewer #2: All comments have been addressed

2. Is the manuscript technically sound, and do the data support the conclusions?

Reviewer #1: Partly

Reviewer #2: Yes

3. Has the statistical analysis been performed appropriately and rigorously? 

Reviewer #1: Yes

Reviewer #2: Yes

4. Have the authors made all data underlying the findings in their manuscript fully available?

Reviewer #1: Yes

Reviewer #2: Yes

5. Is the manuscript presented in an intelligible fashion and written in standard English?

Reviewer #1: Yes

Reviewer #2: Yes

6. Review Comments to the Author

Reviewer #1: The authors have made the necessary corrections and the article is now in better shape. I have no other requests.

Reviewer #2: Thank you for addressing all of my comments. I think it's ready for publishing. I applaud for the great work by the authors.

7. PLOS authors have the option to publish the peer review history of their article (what does this mean?). If published, this will include your full peer review and any attached files.

Reviewer #1: No

Reviewer #2: No

---

## [Editor Report · Acceptance letter]

15 Jul 2024

PONE-D-24-00173R1 

PLOS ONE

Dear Dr. San José Estépar, 

I'm pleased to inform you that your manuscript has been deemed suitable for publication in PLOS ONE. Congratulations! Your manuscript is now being handed over to our production team.

Kind regards, 

on behalf of

Dr. Lorenzo Faggioni 

Academic Editor

PLOS ONE